# Benzalkonium Chloride and Benzethonium Chloride Effectively Reduce Spore Germination of Ginger Soft Rot Pathogens: *Fusarium solani* and *Fusarium oxysporum*

**DOI:** 10.3390/jof10010008

**Published:** 2023-12-22

**Authors:** Dongxu Zhao, Yang Zhang, Zhaoyang Jin, Ruxiao Bai, Jun Wang, Li Wu, Yujian He

**Affiliations:** 1School of Chemical Sciences, University of Chinese Academy of Sciences, Beijing 100049, China; 2Institute of Farmland Water Conservancy and Soil Fertilizers, Xinjiang Academy of Agricultural Reclamation Sciences, Shihezi 832000, China; 3State Key Laboratory of Natural and Biomimetic Drugs, School of Pharmaceutical Sciences, Peking University, Beijing 100191, China; 4School of Future Technology, University of Chinese Academy of Sciences, Beijing 100049, China

**Keywords:** ginger soft rot, *Fusarium solani*, *Fusarium oxysporum*, benzalkonium chloride, benzethonium chloride, inactivation

## Abstract

Ginger soft rot is a serious soil-borne disease caused by *Fusarium solani* and *Fusarium oxysporum*, resulting in reduced crop yields. The application of common chemical fungicides is considered to be an effective method of sterilization, and therefore, they pose a serious threat to the environment and human health due to their high toxicity. Benzalkonium chloride (BAC) and benzethonium chloride (BEC) are two popular quaternary ammonium salts with a wide range of fungicidal effects. In this study, we investigated the fungicidal effects of BAC and BEC on soft rot disease of ginger as alternatives to common chemical fungicides. Two soft rot pathogens of ginger were successfully isolated from diseased ginger by using the spread plate method and sequenced as *F. solani* and *F. oxysporum* using the high-throughput fungal sequencing method. We investigated the fungicidal effects of BAC and BEC on *F. solani* and *F. oxysporum*, and we explored the antifungal mechanisms. Almost complete inactivation of spores of *F. solani* and *F. oxysporum* was observed at 100 mg/L fungicide concentration. Only a small amount of spore regrowth was observed after the inactivation treatment of spores of *F. solani* and *F. oxysporum* in soil, which proved that BAC and BEC have the potential to be used as an alternative to common chemical fungicides for soil disinfection of diseased ginger.

## 1. Introduction

Ginger (*Zingiber officinale Roscoe*) is a significant cash and spice crop in certain tropical and subtropical countries [1,2]. China is the second-largest ginger producer in the world, with about 3 million hectares under ginger cultivation [3], and accounts for about a quarter of the world’s output [4]. Continuous farming causes ginger cultivation to be impacted by a variety of soil-borne diseases, including bacteria and fungi [3]. As a prevalent disease in large ginger soils, the ginger disease soft rot caused by *Fusarium solani* and *Fusarium oxysporum* [5,6,7,8] poses a great threat to the yield and quality of ginger.

Currently, the use of certain chemicals alleviates the condition to some extent, but these agents often come with some potential harmful effects as well. Methyl bromide was once the most widely used soil fumigant in the world and was used for ginger soil fumigation [9], until it was banned globally due to the discovery of stratospheric ozone depletion and lung damage, cytotoxicity, genotoxicity, visual impairment, and various other toxic effects in humans [10,11,12,13]. Chloropicrin, as an alternative to methyl bromide, is an effective alternative fumigant for killing plant pathogens and controlling soil-borne diseases and is widely used for soil disinfection [14,15,16]. Chloropicrin is a highly toxic and potent irritant and tearing compound that was used as a war agent during World War I, causing eye, skin, and respiratory damage [17,18,19]. For this reason, the search for more sustainable and safe control methods for soil disinfection has drawn a lot of attention.

Quaternary ammonium salts are the broad spectral and high effective fungicides, with the advantages of low toxicity, high stability, and high penetration capacity and are widely used in pharmaceutical, chemical, and cosmetic industries, environmental protection, and agriculture [20,21,22,23,24]. Benzalkonium chloride (BAC) and benzethonium chloride (BEC), as two quaternary ammonium salts and surfactants, can adsorb and penetrate into the cell wall, altering the permeability of cytoplasmic membranes and causing intracellular substances to extravasate and degrade, thus acting as a bactericide and fungicide [25,26,27,28,29,30,31]. They have no significant negative effects on soil ecology and have the potential to replace traditional soil fumigants for the treatment of ginger soft rot in soil. The aim of this study was to determine the inactivation efficiencies of spores of *F. solani* and *F. oxysporum* by BAC and BEC.

## 2. Materials and Methods

### 2.1. Isolation and Characterization of Pathogenic Fungi

The isolation and identification of pathogenic fungi was slightly improved compared to previous methods [32,33,34]. Diseased ginger samples were taken from the field in Anqiu, Shandong Province (36.48 N, 119.22 E), taken to the laboratory in sealed bags, and stored in the refrigerator at 4 °C. Diseased ginger surfaces were gently scrubbed with a small brush, cut into 2 to 3 mm cubes, surface sterilized by immersion in 75% ethanol for 30 s, rinsed three times with sterile water, and dried on sterile filter paper for 1 min. A total of 2 g of diseased ginger mixture was taken in a 15 mL centrifuge tube, gently mashed by a sterile glass rod, 5 mL of sterile water was added, and the mixture was shaken well. The mixture was spread on chloramphenicol-containing potato dextrose agar (PDA, containing 0.1 g/L chloramphenicol, Guangdong Huankai Microbial Technology Co., Ltd., Guangzhou, China) plates, and incubated at a constant temperature of 28 °C for 5 days. Then, fungal colonies were selected and incubated again on the PDA [32,33,34]. Fungal ITS was performed on the purified colonies for identification of the fungi.

The isolated colonies were subjected to genome extraction by proteinase K digestion and lysis and recovered using the AxyPrep DNA Gel Recovery Kit (AP-GX-250, AXYGEN Biosciences, Union City, CA, USA). The two separated samples were subjected to high-throughput sequencing, the internal transcribed spacer (ITS) region 1 of the fungi was amplified using the ITS1F (5′-CTTGGTCATTTAGAGGAAGTAA-3′) and ITS2R (5′- TCCTCCGCTTATTGATATGC-3′) primer pair. Multiple sequence alignment alignments were performed using MUSCLE (version: 3.28.0) software and developmental trees (Bootstrap = 1000) were constructed by Neighbor-Joining with MegaCC (version: 10.1.8) software and visualized using Ggtree (version: 2.0.4) (TinyGene Bio-Tech (Shanghai) Co., Ltd., Shanghai, China).

### 2.2. Preparation of Spore Suspensions

PDA was used to culture the fungi in a constant temperature incubator at 28 °C for 5–7 days [35]. After that, fungal spores were rinsed from the PDA plates into 50 mL centrifuge tubes with sterile PBS, and then, spores and mycelium were separated by passing through a 300-mesh cell sieve. Finally, the fungal spores were further purified by centrifugation (3000 rpm, 5 min) [35,36]. The initial concentration of fungal spores was observed as 10^7^–10^8^ CFU/mL under a 400× inverted fluorescence microscope (DM500, Leica, Wetzlar, Germany) using a hemocytometer [37].

### 2.3. Chloropicrin Soil Sterilization Experiment

Sterilized centrifuge tubes (50 mL, NUNC, Rochester, NY, USA) containing 10 g of soil (taken from Anqiu, Shandong) were sterilized in an autoclave at 121 °C for 60 min. The spore suspensions of *F. solani* and *F. oxysporum* were diluted to 10^6^ CFU/mL, 1 mL of spore suspension was pipetted into the centrifuge tubes using a 1000 µL pipettor and stirred repeatedly with a pipette tip, which was left inside the centrifuge tube in the end. Chloropicrin (99.5%, Shandong Anqiu Farmers’ Market, Weifang, China) was added to centrifuge tubes at one to four times the field application rate (50, 100, 150, 200 mg/kg), and the tubes were sealed and continued to incubate at 28 °C for 2 days. A total of 10 mL of sterile water was added to the centrifuge tube, and after shaking well, 100 µL of the suspension was spread on a PDA plate, and incubated at a constant temperature of 28 °C for 7 days [14,15]. Each sample was taken in triplicate.

### 2.4. Colony Growth Inhibition of F. solani and F. oxysporum by BAC and BEC

After autoclaving, the PDA medium was cooled to about 60 °C, and BAC (99.7%, Shanghai Dingxian Biotechnology Co., Ltd., Shanghai, China) and BEC (50% in water, Shanghai Jizi Biochemistry Co., Ltd., Shanghai, China) were added to make the final concentration of the solution at 10 mg/L, 50 mg/L, 100 mg/L, 500 mg/L, 1000 mg/L, shaken well, and transferred evenly into Petri dishes in triplicate and allowed to solidify. Three plates without compounds were used as controls. PDA plates uniformly covered with *F. solani* and *F. oxysporum* were perforated using a 6 mm perforator to make uniformly shaped fungus pieces, which were inoculated face up in the center of the compound-containing medium and control medium, respectively. All Petri dishes were incubated at a constant temperature of 28 °C for 7 days and the radial growth of the colonies was measured, at which time the area of fungal hyphae on the control plate just covered the Petri dish. Plate inhibition was calculated using the following formula [38,39]:(1)$$colony\;diameter\;\left(mm\right)=measurement\;of\;mean\;colony\;diameter-6\;mm,$$
(2)$$Mycelial\;growth\;inhibition\;rate\;\left(\%\right)=\frac{control\;colony\;diameter-treatment\;colony\;diameter}{control\;colony\;diameter}\times 100\%$$
where 6 mm denotes the diameter of mycelium when freshly inoculated on the medium.

### 2.5. Spore Inactivation Experiment

Inactivation experiments using both BAC and BEC were performed in centrifuge tubes (50 mL, NUNC, Rochester, NY, USA). For inactivation experiments, concentrated inoculum was prepared as described in Section 2.3. Spore suspensions were diluted using PBS to prepare a final concentration of 10^6^ spores/mL. The inactivation experiment was initiated by mixing 100 times the desired concentration of BAC and BEC solution into the spore suspension at a volume ratio of 1:100, followed by thorough stirring. At the end of the inactivation experiment, the spores were separated from the solution containing BAC and BEC by centrifugation (3000 rpm, 5 min), the upper layer of solution was removed, and the spores were resuspended in PBS [40,41]. Subsequent experimental procedures were the same as described above for individual inactivation. Sterile water was used instead of the compound for control experiments.

To investigate the effect of compound concentration on inactivation efficiency, six different initial compound concentrations (10, 50, and 100 mg/L of BAC and BEC) were selected for inactivation experiments on fungal spores. The inactivation time of each mixture was controlled at 1, 5, 10, 15, 30, and 60 min to investigate the effect of inactivation time on inactivation efficiency [42,43].

### 2.6. Methods Used for Observation of Changes in Spore Activity

#### 2.6.1. Laser Confocal Microscopy Observation

The activity of two fungal spores was observed using laser confocal microscopy (DMI 8, Leica, Wetzlar, Germany) combined with two selective dyes. Propidium iodide (PI, 50% in water, Shanghai Jizi Biochemistry Co., Ltd., Shanghai, China)/fluorescein diacetate (FDA, Shenzhen Baitaike Technology Co., Ltd., Shenzhen, China) combination staining was chosen to rate the percentage of intact and cellular damage. FDA is a nonfluorescent and cell-permeable viability probe that is de-esterified by cytosolic esterases and converted to the highly fluorescent compound fluorescein in living cells [44], whereas propidium iodide is a cell-impermeable nucleic acid stain that is internalized only when membranes are damaged [45]. Briefly, FDA (dissolved in Dimethyl sulfoxide, DMSO, Shanghai McLean Biochemical Technology Co., Ltd., Shanghai, China) and PI (dissolved in ddH_2_O) were added to the treated spore suspensions at final concentrations of 10 µg/mL and 5 µg/mL, respectively, to make the samples homogeneous for staining using vortex shaking, and the samples were incubated using a constant-temperature metal bath in the dark at 28 °C for 20 min. Laser confocal microscopy was used to capture photographic images of the stained spores. The excitation/emission wavelengths were 488/530 nm for FDA and 535/615 nm for PI.

#### 2.6.2. Flow Cytometry Analysis

Changes in the activity indices of spores of two fungal species during the inactivation procedure were determined by flow cytometry combined with PI [46]. At the end of the inactivation experiment, PI was added to the sample centrifuge tubes at a final concentration of 5 µg/mL, and the samples were stained homogeneously using vortex shaking, and incubated for 20 min at 28 °C in the dark using a thermostatic metal bath. Finally, the sample concentration was diluted to one-tenth of the original, and the diluted sample was analyzed by flow cytometry. Red fluorescence emitted by the PI was detected at ~670 nm (FL3 channel), and 10,000 events were detected per sample. The data were collected and analyzed using FlowJow software version 10.

#### 2.6.3. Scanning Electron Microscopy (SEM) Morphological Observations

Sample preparation was based on previous studies [47,48] with some modifications. To detect changes in spore morphology, fungal spore suspensions from the experimental and control groups were first centrifuged at 10,000 rpm for 15 min, the supernatant was discarded, and then, 3% glutaraldehyde was added and stored in a refrigerator at 4 °C overnight. The spores were rinsed with graded concentrations of ethanol (10%, 30%, 50%, 70%, 90%, and 100%) to remove the excess glutaraldehyde, and then, 1 mL of isoamyl acetate were added and shaken well, and then, silicon wafers on aluminum stubs were used to mount samples, which were naturally dried and then sprayed with gold on the surface, and the morphological characteristics of spores were observed by a scanning electron microscope (SEM; SU8010, HITACHI, Tokyo, Japan) at 5000/10,000× magnification.

#### 2.6.4. Determination of Regrowth Potential

The regrowth potential of the spores after inactivation was first evaluated in the potato dextrose broth (PDB, Beijing Solebao Technology Co., Ltd., Beijing, China). The regrowth potential of BAC and BEC treatments was assessed using the 96-well microtiter plate method. First, PDB liquid medium was mixed with fungal spore suspension in 50 mL centrifuge tubes so that the spore concentration was controlled at 10^6^ spores/mL, and BAC and BEC were added to make the medium fungicidal concentrations of 0, 10, 50, and 100 mg/L [49]. Every four hours, 200 µL of solution was added to one well of a 96-well plate, and the operation was repeated six times to reduce experimental error. The change in optical density at 600 nm (OD_600_) [50] was detected using a microplate reader (Spark, Tecan, Männedorf, Switzerland) [51].

In addition, we inoculated the treated spore suspension directly into soil to observe its regrowth potential in soil. This was carried out by adding 1 mL of 10^6^ CFU/mL spore suspension to a 50 mL centrifuge tube containing 10 g of heat-sterilized soil (36.48° N, 119.22° E, pH 7.54, EC 132.5 µS/cm, water content 15%) to reach a spore count of about 10^6^ spores in the centrifuge tube, mixing thoroughly with a 1000 µL pipette tip. Then, add 10 mL of BAC/BEC at a concentration of 100 mg/L (10 mL of sterile water for the control), seal the centrifuge tube, and shake it in a water bath shaker for 30 min. The upper solution in the centrifuge tube was then diluted 10-fold with sterile water, and 100 µL of each solution was placed in PDA plates and incubated in the dark for 2 days to observe the phenotype. At this time, the theoretical number of spores on the plate was about 1000.

The inactivation efficiency was calculated with the following formula:(3)$$S=(N_{0}-N_{t})/N_{0}$$
where *S* denotes the inactivation efficiency, *N_t_* denotes the number of viable fungal spores at time *t*, and *N*_0_ denotes the number of viable fungal spores at the beginning of the experiment.

## 3. Results and Discussion

### 3.1. Identification of Pathogenic Fungi in Ginger

To identify and isolate the pathogenic fungi from ginger, 21 soil samples from nine different areas and more than 10 samples of diseased ginger root were taken from the Anqiu Country of Shandong Province. From these samples, several samples with obvious characteristics were selected for the mixed culture, and multiple strains of the two fungi were isolated using PCR amplification and fungal ITS heavy region sequencing, and two strains of fungal pathogens were selected as experimental samples (as shown in the phylogenetic tree in Figure 1) [5,6,7,8]. The two pathogens of ginger stem rot, *F. solani* (Figure 2a) and *F. oxysporum* (Figure 2b), were finally identified.

### 3.2. Antifungal Effect of Chloropicrin in Soil

The effects of chloropicrin fumigation on *F. solani* and *F. oxysporum* in the soil were simulated in centrifuge tubes. As shown in Figure 3, the results of the plate culture showed that the amount of chloropicrin (50 mg/kg), which was commonly used by farmers for soil disinfection in the field, was not effective in inactivating these two fungi. *F. solani* spores from soil fumigated with 100 mg/kg as they remained viable in plate cultures; this concentration was well in excess of that causing sensory irritation and respiratory damage in humans at 8 mg/kg [52]. This result suggested that fumigation with low concentrations of chloropicrin was not effective in inactivating the soft rot pathogen of ginger in the soil.

### 3.3. Mycelial Growth Inhibitory Concentration of BAC and BEC

The effective inhibitory concentrations were initially explored in PDA plates containing different BAC and BEC concentrations (10, 50, 100, 500, 1000, and 5000 mg/L), as shown in Figure 4. For *F. solani* (Figure 4a), with the low concentration of BAC, it was difficult to produce an effective and complete inactivation effect, and a larger area of growth compared with the control was still produced (mycelial growth in PDA medium containing different concentrations of compounds is shown in Appendix A), and the inhibition rate was only 56% when the concentration was increased to 100 mg/L. When the inactivation concentration continued to increase, the mycelium stopped growing. BEC at the same concentration produced a better inhibitory effect on *F. solani*, at the concentration of 100 mg/L, the inhibition rate of *F. solani* reached 89%, which was better than that of BAC in terms of a fungicidal effect (*p* < 0.01). For *F. oxysporum* (Figure 4b), the plate inhibition effect of BEC was more obvious; 10 mg/L of BEC could produce a 57% inhibition effect, while the inhibition rate of BAC at the same concentration was only 41%. With the increase in fungicidal concentration, the advantage of BEC was more obvious (*p* < 0.01). When the fungicidal concentration was raised to 500 mg/L, a small amount of mycelium growth was still produced on the plate under the BAC treatment condition, while mycelium growth on the plate under the BEC treatment condition stopped completely, which also indicated that the inhibition of mycelium by BEC was more efficient.

### 3.4. Changes in Spore Activity

In order to determine the effect of time and concentration on the post-inactivation spore activity of BAC and BEC, two well-known dyes, the membrane-impermeable agent PI and the membrane-permeable agent FDA, were used. The fluorescence changes of the spores after the different reaction conditions were observed using a laser confocal microscope, as shown in Figure 5. The majority of spores from the negative control fluoresced green, which was caused by fluorescent luciferin produced by the cytosolic esterase upon cleavage by FDA, indicating that they were viable. The proportion of positive PI staining increased significantly with increasing fungicidal concentration (as shown in Appendix A, where spore viability is expressed as the ratio of red fluorescence intensity to the total fluorescence intensity). When the fungicidal concentration was increased to 50 mg/L, nearly half of the spores were stained to show red fluorescence, and when the fungicidal concentration was increased to 100 mg/L, for spores of *F. solani* and *F. oxysporum*, the majority of the spores showed strong PI red fluorescence, indicating that they were dead cells with disrupted membranes.

The inactivation of spores of *F. solani* and *F. oxysporum* by fungicidal time and concentration was determined using flow cytometry in combination with PI (Figure 6). The inactivation effect was shown in Figure 7, and it can be found that the BAC and BEC have almost completed the inactivation of spores in a very short period of time (30 s) and stabilized after 5 min. For *F. solani* spores (Figure 7a), both antifungal compounds showed a good inactivation effect, under the application of 50 mg/L concentration, the inactivation of both compounds against fungal spores reached 99.8% in 30 s, and under the condition of 10 mg/L benzalkonium chloride concentration, the inactivation also reached 89.7% after 30 min of treatment. For the *F. oxysporum* spores (Figure 7b), different concentrations of the two fungicides produced a clear inactivation gradient (Appendix A), and overall, the antifungal effect of BAC was better than that of BEC. In total, 56.1% of spores were inactivated by BAC in 1 min at a concentration of 10 mg/L, and the inactivation rate was stabilized over a longer period of time. The inactivation rate of 50 mg/L BAC at 30 min was 92.8%, which was already higher than that of 100 mg/L BEC (89.4%), and the inactivation rate of 100 mg/L BAC was close to 100%. The trend of the curves in Figure 7 shows that the inactivation of spores was concentrated in 1 min or less, and the inactivation rate of spores tended to stabilize with the extension of time. This result was different from the experimental results in Section 3.3, a phenomenon described as “viable but non-culturable” (VBNC) [53].

### 3.5. Changes in Spore Surface Morphology and Structure

The morphological changes of *F. solani* and *F. oxysporum* after applying the BAC and BEC were observed using SEM. As shown in Figure 8, the surface of *F. solani* spores (a) and *F. oxysporum* spores (b) before inactivation were smooth and intact, with a slightly curved sickle-shaped morphology. *F. solani* spores with a length of about 10 µm became rough, wrinkled, and sunken with a shriveled surface after inactivation with 100 mg/L of BAC and BEC. For *F. oxysporum* spores, there was no significant change in spore length before and after inactivation, which was about 5 µm, and the surface of the BAC-inactivated spores was more severely wrinkled, while the surface of the BEC-inactivated spores was relatively intact, which was consistent with the findings in Section 3.4. Gang, W. et al. [53] conducted a study on the spore inactivation of *Penicillium polonicum*, *Trichoderma harzianum*, and *Cladosporium cladosporioides*. The SEM characterization of the inactivated spores revealed that the surface of the inactivated spores became rough and wrinkled, the cell membranes and cell walls were disrupted, and a small number of compounds were released around the spores. Zuo, J. et al. [47,48,54,55] performed SEM characterization of various spores of Aspergillus spp. and similarly observed morphological damage. The SEM characterization results in this study correspond to previous studies, confirming that the cell membranes and cell walls of both spores were disrupted, leading to the release of intracellular compounds that inactivate the fungal spores. BAC inactivated both fungal spores relatively well compared to BEC.

### 3.6. Regrowth Potential after BAC/BEC Treatment of Spores

As shown in Figure 9, both spores of *F. solani* and *F. oxysporum* exhibited significant regrowth potential before inactivation treatments with BAC and BEC. For *F. solani*, as shown in Figure 9a, BAC and BEC at 10 mg/L were unable to effectively inactivate the spores, resulting in a spore regrowth trend similar to that of the control, which produced a logarithmic proliferation after 8 h. When the compound concentration was increased to 50 mg/L, no regrowth of *F. solani* was observed even though the inactivation time was shortened to 15 min (Figure 9b). The inactivation of *F. oxysporum* spores was different from those of *F. solani*. In Figure 9a, after inactivation with 10 mg/L of BEC, spore regrowth lagged significantly and began to grow exponentially after 12 h. (Figure 9c). As shown in Figure 9d, spore regrowth was also observed 48 h after inactivation with 50 mg/L of BEC for different periods of time, which demonstrated that spore inactivation occurs in a shorter period of time, and that the inactivation of *F. oxysporum* spores by BEC was inferior to that by BAC, which corresponds to the results of the previous study.

In addition, the effect of BAC and BEC on the inactivation of spores of both fungi was also investigated under soil conditions, and the regrowth of spores was observed by plate coating. As shown in Figure 10, the spores regenerated after inactivation by BAC and BEC compared with the number of spores in the control group, but the inactivation rates of *F. solani* spores (Figure 10a) and *F. oxysporum* spores (Figure 10b)) were still 90.7% and 98% and 97.9% and 99.6%, which proved that these two compounds still have a good effect of spore inactivation in the soil and have the potential to be applied to soil inactivation of fungi spores. Direct application of BAC and BEC to soil containing the pathogen will be an effective method of inhibiting the spread of ginger soft rot pathogens.

### 3.7. Discussion

In this study, two soft rot causing fungi, *F. solani* and *F. oxysporum*, were successfully isolated from ginger samples. The inactivation of *F. solani* and *F. oxysporum* proved that lower concentrations of BAC and BEC produced effective inactivation, which may have the potential to be applied to the inactivation of other soil pathogens. The inactivation of spores was proven to be the inactivation of intracellular material due to osmosis, rather than disruption of the cell membrane, by observing the integrity of the spore membrane and changes in cell morphology. The measurement of the regrowth potential of spores after inactivation treatment demonstrated that the inactivation of spores by BAC and BEC was effective and irreversible, but more importantly, the effect of fungicides on the disinfection of *F. solani* and *F. oxysporum* in soil was assessed.

When chloropicrin is photolyzed, the photolysis products phosgene, nitrosyl chloride, chloride, and nitric oxide are inhalation hazards and are classified as Toxicity Category 1. Numerous countries have imposed restrictions on their use due to the lack of therapeutic treatments for chloropicrin exposure [18,19]. In contrast, BAC and BEC exhibit superior stability, resist spontaneous degradation, and possess significantly lower toxicity than chloropicrin. However, the enhanced stability and adsorption properties of BAC and BEC increase their retention in soil layers, reducing their likelihood of runoff migration [56]. This can lead to environmental complications. Notably, several studies have successfully identified microorganisms capable of degrading quaternary ammonium salts from marine sediments, including the Gram-negative bacterium *Thalassospira sp.* and the Gram-positive bacterium *Bacillus niabensis* [57]. Cui, Y. et al. [58] extracted *Klebsiella*, *Enterobacter*, *Citrobacter*, and *Pseudomonas* species, which facilitated the degradation of BEC. Furthermore, the efficacy of advanced oxidation processes in decomposing quaternary ammonium salts presents a promising solution [59].

The use of BAC and BEC is also controversial in some regions. In Europe, the European Commission is involved in the regulation of BAC, which is no longer approved for use in several biocide products such as consumer hand sanitizers and body wash preservatives, unlike current legislation in the United States. The U.S. Food and Drug Administration recently published three proposed and final decisions on the use of chemicals other than BAC and BEC as preservatives in consumer hand soaps, consumer hand and body washes, and health care [26]. Therefore, the use of BAC and BEC in soils is also subject to further study.

## 4. Conclusions

In summary, two quaternary ammonium salts, BAC and BEC, in the treatment of the soft rot pathogens *F. solani* and *F. oxysporum* in ginger were investigated as alternatives to the soil fumigant chloropicrin. Spore inactivation experiments demonstrated that BAC and BEC were highly effective in inactivating spores of *F. solani* and *F. oxysporum*. The inactivation of spores originated from permeation rather than disruption of the cell membrane by morphological analysis. The regrowth of spores after BAC and BEC treatments was studied in soil, which demonstrated that BAC and BEC have the potential to be applied as an alternative to chloropicrin in the treatment of ginger soft rot. And direct application of BAC and BEC to soil containing the pathogen would be a promising method to inhibit the spread of ginger soft rot pathogens.

## Figures and Tables

**Figure 1 jof-10-00008-f001:**
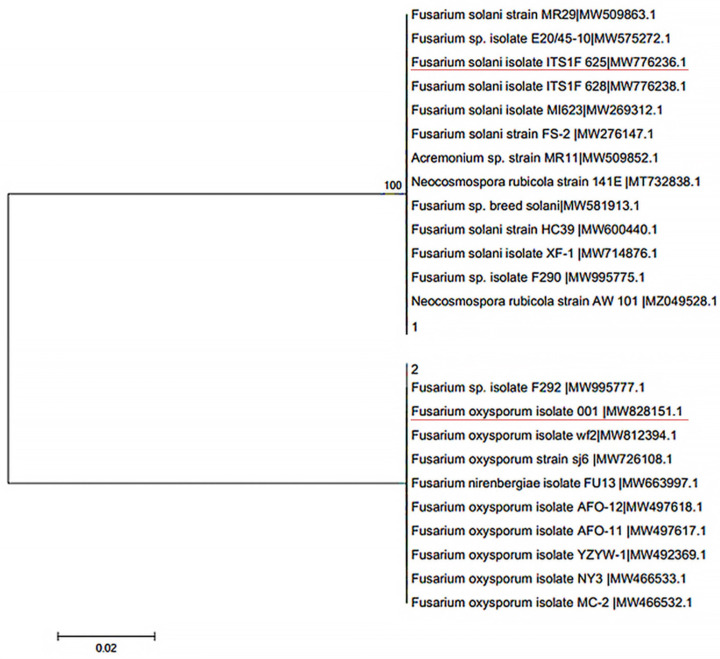
Phylogenetic tree of fungal identification. The two strains of fungal pathogens used as experimental samples were underlined in red.

**Figure 2 jof-10-00008-f002:**
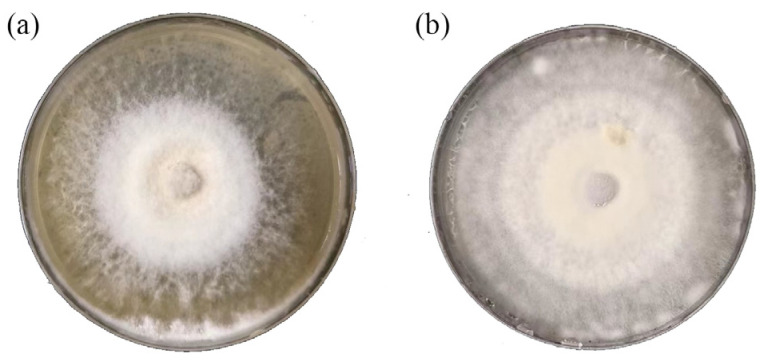
After 7 days of incubation on PDA medium, two fungi were isolated from diseased ginger root tissues. (**a**) *Fusarium solani*, (**b**) *Fusarium oxysporum*. Incubation conditions: pH = 7.40; T = 28 ± 1 °C.

**Figure 3 jof-10-00008-f003:**
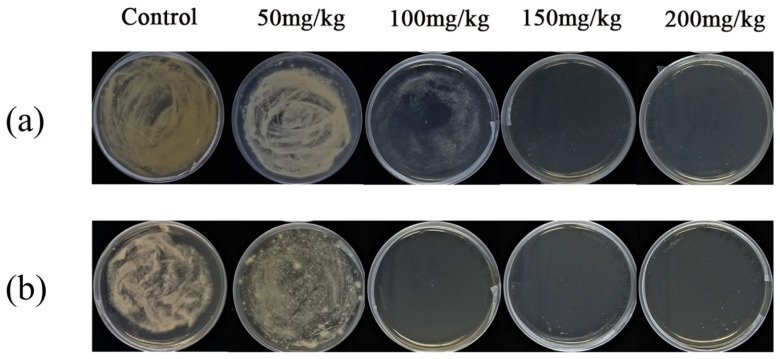
Effects of chloropicrin fumigation in the soil on *Fusarium solani* (**a**) and *Fusarium oxysporum* (**b**). Experimental conditions: [chloropicrin] = 50, 100, 150, 200 mg/kg; fungal spores = 10^6^ CFU/mL; pH = 7.40; T = 28 ± 1 °C.

**Figure 4 jof-10-00008-f004:**
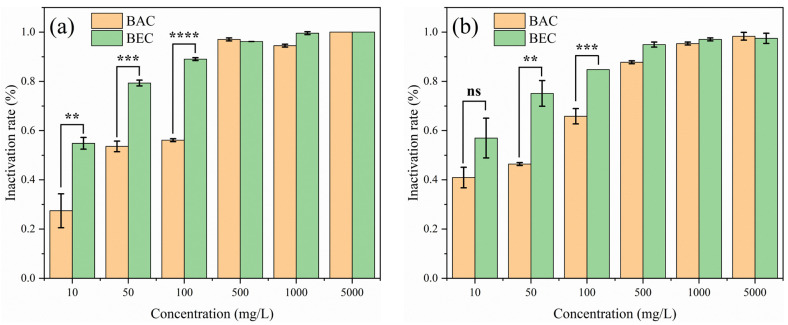
Comparison of mycelial growth of *Fusarium solani* (**a**) and *Fusarium oxysporum* (**b**) after 7 days of incubation on BAC- and BEC-containing plates. Experimental conditions: [BAC] = [BEC] = 10, 50, 100, 500, 1000, 5000 mg/L; initial concentration of fungal spores = 10^6^ CFU/mL; pH = 7.40; T = 28 ± 1 °C. (*n* = 3, ns for not significant, ** *p* < 0.01, *** *p* < 0.001, **** *p* < 0.0001).

**Figure 5 jof-10-00008-f005:**
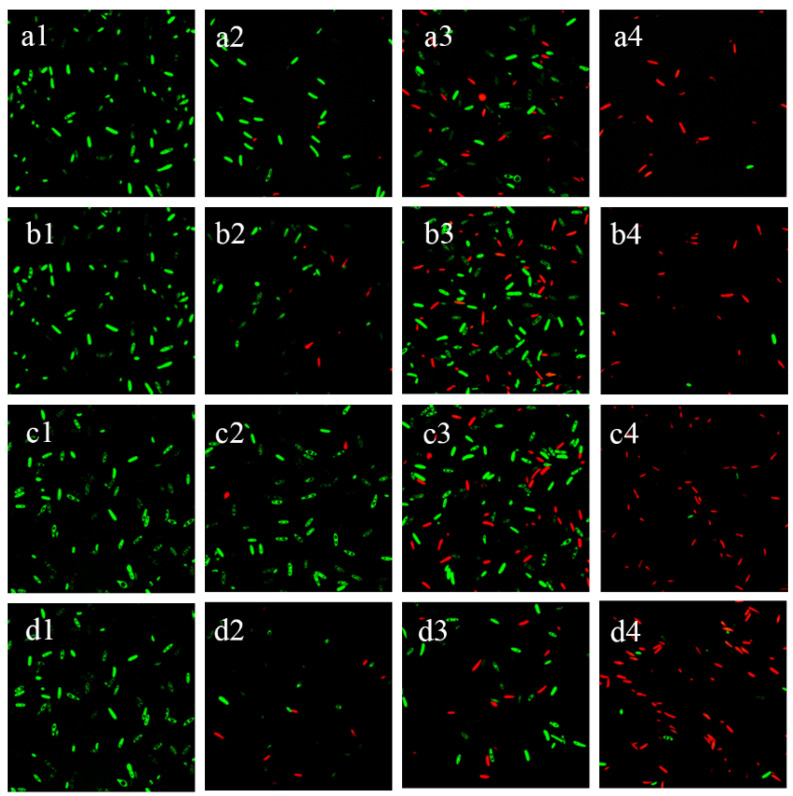
Viability of *Fusarium solani* and *Fusarium oxysporum* analyzed by double staining with fluorescein diacetate (FDA, green fluorescence) and propidium iodide (PI, red fluorescence). (**a1**–**a4**) represent the treatment of *Fusarium solani* spores with 0, 10, 50, and 100 mg/L of BAC for 30 min, respectively; (**b1**–**b4**) represent the treatment of *Fusarium solani* spores with 0, 10, 50, and 100 mg/L of BEC for 30 min, respectively; (**c1**–**c4**) represent the treatment of *Fusarium oxysporum* spores with 0, 10, 50, and 100 mg/L of BAC for 30 min, respectively; and (**d1**–**d4**) represent the treatment of *Fusarium oxysporum* spores with 0, 10, 50, and 100 mg/L of BEC for 30 min, respectively. Experimental conditions: initial concentration of fungal spores = 10^6^ CFU/mL; pH = 7.40; T = 28 ± 1 °C.

**Figure 6 jof-10-00008-f006:**
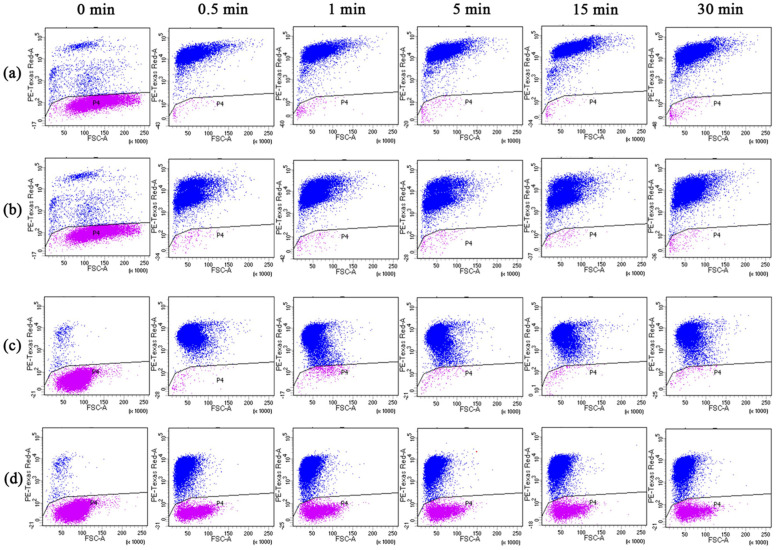
Two-dimensional dot plots of *Fusarium solani* and *Fusarium oxysporum* analyzed using flow cytometry: (**a**) BAC treatment of *Fusarium solani*; (**b**) BEC treatment of *Fusarium solani*; (**c**) BAC treatment of *Fusarium oxysporum*; and (**d**) BEC treatment of *Fusarium oxysporum*. Where purple represents FDA staining results and blue represents PI staining results. Experimental conditions: [BAC] = [BEC] = 100 mg/L; initial concentration of fungal spores = 10^6^ CFU/mL; pH = 7.40; T = 28 ± 1 °C.

**Figure 7 jof-10-00008-f007:**
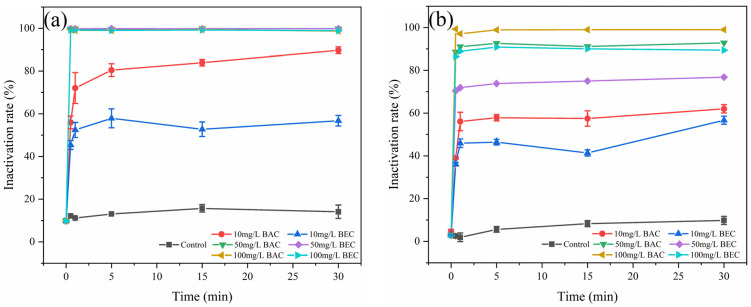
Flow cytometry analysis of the inactivation process of two fungal spores using PI. (**a**) *Fusarium solani* spores; (**b**) *Fusarium oxysporum* spores. Experimental conditions: [BAC] = [BEC] = 10, 50, 100 mg/L; sterile water was used instead of compound for control experiments; initial concentration of fungal spores = 10^6^ CFU/mL; pH = 7.40; T = 28 ± 1 °C.

**Figure 8 jof-10-00008-f008:**
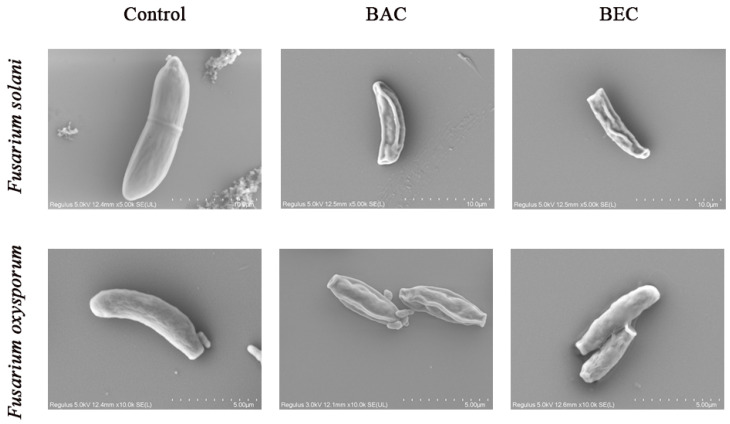
SEM images of fungal spores after inactivation using BAC and BEC. *Fusarium solani* spores were magnified 5000 times and *Fusarium oxysporum* spores were magnified 10,000 times. Experimental conditions: [BAC] = [BEC] = 100 mg/L; initial concentration of fungal spores = 10^6^ CFU/mL; pH = 7.40; T = 28 ± 1 °C; reaction time= 30 min.

**Figure 9 jof-10-00008-f009:**
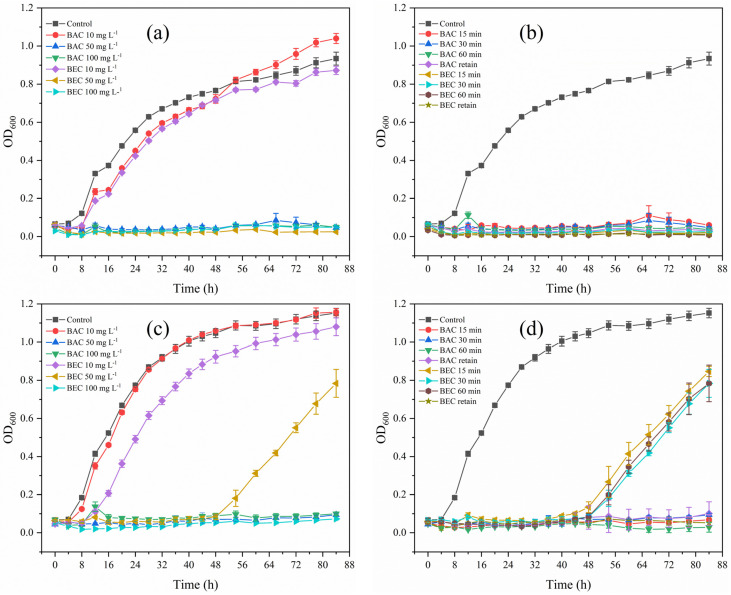
Regrowth of spores of *Fusarium solani* and *Fusarium oxysporum* in PDB after treatment with BAC/BEC. (**a**) *Fusarium solani* spores were treated with different concentrations of BAC and BEC for 30 min; (**b**) *Fusarium solani* spores were treated with 50 mg/L of BAC and BEC for different times; (**c**) *Fusarium oxysporum* spores were treated with different concentrations of BAC and BEC for 30 min; (**d**) *Fusarium oxysporum* spores were treated with 50 mg/L of BAC and BEC for different times. Experimental conditions: initial concentration of fungal spores = 10^6^ CFU/mL; pH = 7.40; T = 28 ± 1 °C.

**Figure 10 jof-10-00008-f010:**
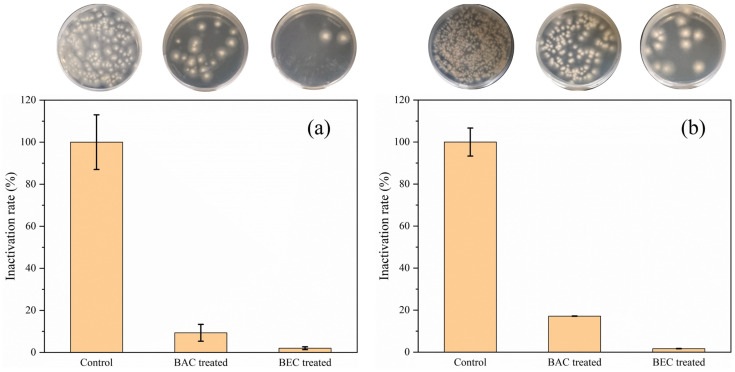
Effect of BAC and BEC on the inactivation of *Fusarium solani* spores (**a**) and *Fusarium oxysporum* spores (**b**) in sterilized soil samples. Experimental conditions: [BAC] = [BEC] = 100 mg/L; initial concentration of fungal spores = 10^6^ CFU/mL; pH = 7.40; T = 28 ± 1 °C; reaction time = 30 min.

## Data Availability

Data are contained within the article and Appendix A.

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
