# Peer review of "Benzalkonium Chloride and Benzethonium Chloride Effectively Reduce Spore Germination of Ginger Soft Rot Pathogens: *Fusarium solani* and *Fusarium oxysporum"

_jof, 2023, doi:10.3390/jof10010008_

Round 1

Reviewer 1 Report (Previous Reviewer 3)

Comments and Suggestions for Authors

Put the Latin phrase "in vitro" in italic in Reference section, Lines 454, 464, 497, 500.

Author Response

Benzalkonium chloride and benzethonium chloride effectively reduce spore germination of ginger soft rot pathogens: Fusarium solani and Fusarium oxysporum

Response to Reviewer 1 Comments

1. Summary

2. Questions for General Evaluation

Reviewer’s Evaluation

Response and Revisions

Does the introduction provide sufficient background and include all relevant references?

Yes

Are all the cited references relevant to the research?

Yes

Is the research design appropriate?

Yes

Are the methods adequately described?

Yes

Are the results clearly presented?

Yes

Are the conclusions supported by the results?

Yes

3. Point-by-point response to Comments and Suggestions for Authors

  • Put the Latin phrase "in vitro" in italic in Reference section, Lines 454, 464, 497, 500.

Response: Thank you for your advice. Relevant content has been italicized. Lines 459, 469, 502, 505.

Reviewer 2 Report (Previous Reviewer 1)

Comments and Suggestions for Authors

1) Do you propose BAC and BEC as a mordants? coating the ginger roots before planting? Or do you propose use BEA and BAC as a soil preparation for direct soil application, soil disinfectant? This could be specified in the introduction. Or maybe in the discussion or conclusion.

2) L 120: Change to the Spore inactivation soil experiment...to be consistent with Chlorpicrin experiment

3) L 137: Change title to Methods used for observation of changes in spore activity.

4) L 227:  This chapter title seems to be a nonsense. Replace it with e.g. Mycelial growth inhibitory concentration of BAC and BEC

5) L 290: Make it consistent with the renamed title of assay recomended higher. Do not use "inactivation of the compound-containing plate"

6) L 297: Figure 7 description: Add an information what was used as a control (water, or water with some detergent or water with some amount of solvent used for BAC and BEC solvatation in other variants?) The same information give in the methods chapter.

7) L 329: Do not use a disinfection concentration term. Better use inhibitory concentration or just compounds concentration.

8) L349 - 350: change the sentence to ...which proved that these two compounds still have good effect of spore inactivation in the soil and have the potential.... etc.  

Author Response

Benzalkonium chloride and benzethonium chloride effectively reduce spore germination of ginger soft rot pathogens: Fusarium solani and Fusarium oxysporum

Response to Reviewer 2 Comments

1. Summary

2. Questions for General Evaluation

Reviewer’s Evaluation

Response and Revisions

Does the introduction provide sufficient background and include all relevant references?

Can be improved

This section has been improved.

Are all the cited references relevant to the research?

Can be improved

This section has been improved.

Is the research design appropriate?

Can be improved

This section has been improved.

Are the methods adequately described?

Yes

Are the results clearly presented?

Can be improved

This section has been improved.

Are the conclusions supported by the results?

Can be improved

This section has been improved.

3. Point-by-point response to Comments and Suggestions for Authors

  • Do you propose BAC and BEC as a mordants? coating the ginger roots before planting? Or do you propose use BEA and BAC as a soil preparation for direct soil application, soil disinfectant? This could be specified in the introduction. Or maybe in the discussion or conclusion.

Response: Thank you for your valuable suggestions. We have added corresponding content in section 3.6 and conclusion. L 352-353, L 398-400.

“Direct application of BAC and BEC to soil containing the pathogen would be promising methods to inhibit the spread of ginger soft rot pathogens.”

  • L 120: Change to the Spore inactivation soil ..to be consistent with Chlorpicrin experiment.

Response: Thank you for your advice. This section is more concerned with the inactivation of spores in solution, so we don’t think that the title " Spore inactivation soil experiment " is appropriate here. L 119.

  • L 137: Change title to Methods used for observation of changes in spore activity.

Response: Thank you for your advice. The title has been changed to " Methods used for observation of changes in spore activity". L 136.

  • L 227:  This chapter title seems to be a nonsense. Replace it with e.g. Mycelial growth inhibitory concentration of BAC and BEC

Response: Thank you for pointing out the problem. The title has been changed to " Mycelial growth inhibitory concentration of BAC and BEC " as you suggested. L 227.

  • L 290: Make it consistent with the renamed title of assay recomended higher. Do not use "inactivation of the compound-containing plate"

Response: Thank you for your advice. The relevant part has been changed to " This result was different from the experimental results in Section 3.3". L 290.

  • L 297: Figure 7 description: Add an information what was used as a control (water, or water with some detergent or water with some amount of solvent used for BAC and BEC solvatation in other variants?) The same information give in the methods chapter.

Response: Thank you for your comments. Related information has been added. Relevant references are mentioned in the methodology section. L 299, L 129.

“sterile water was used instead of compound for control experiments”.

  • L 329: Do not use a disinfection concentration Better use inhibitory concentration or just compounds concentration.

Response: Thank you for your advice. We have changed " disinfection concentration " to " compounds concentration". L 330.

  • L349 - 350: change the sentence to ...which proved that these two compounds still have good effect of spore inactivation in the soil and have the potential.... etc.  

Response: Thank you for your suggestion. We have changed the sentence as you suggested. “which proved that these two compounds still have good effect of spore inactivation in the soil and have the potential to be applied to soil inactivation of fungi spores.” L 350 - 352.

Reviewer 3 Report (New Reviewer)

Comments and Suggestions for Authors

The article may be of interest for the reader because it shows the efficiency of a treatment against the Fusarium species considered. Anyway, what is shown in the article is not a novelty, seen that the antimicrobial activity of quaternary ammonium salts is already known. According to the GHS (Globally Harmonized System) hazard statements the two quaternary ammonium salts are: H410 - "Very toxic to aquatic life with long lasting effects". Therefore the author's conclusion stating that this has the potential to be used as an alternative to common chemical fungicides for soil disinfection seams pretty unthinkable.
Anyway is better for the authors to take in consideration some minor adjustments:
Line 13 - 14 The scientific name for the two Fusariums needs to be stated only fully.
Line 32 Is better to also state the scientific name of the ginger plant.
Line 37 - 38 The scientific name for the two Fusariums needs to be stated only fully.
Shortly after line 117 I suggest to make the unit measure in the formula in mm as in the text.
Line 176 Maybe is better to explain the acronym the first citation.
Line 373 Explain the acronym QAC.
In References the reference 11 and 12 are not cited anywhere in the text.

Author Response

Benzalkonium chloride and benzethonium chloride effectively reduce spore germination of ginger soft rot pathogens: Fusarium solani and Fusarium oxysporum

Response to Reviewer 3 Comments

1. Summary

2. Questions for General Evaluation

Reviewer’s Evaluation

Response and Revisions

Does the introduction provide sufficient background and include all relevant references?

Yes

Are all the cited references relevant to the research?

Can be improved

This section has been improved.

Is the research design appropriate?

Yes

Are the methods adequately described?

Yes

Are the results clearly presented?

Yes

Are the conclusions supported by the results?

Yes

3. Point-by-point response to Comments and Suggestions for Authors

The article may be of interest for the reader because it shows the efficiency of a treatment against the Fusarium species considered. Anyway, what is shown in the article is not a novelty, seen that the antimicrobial activity of quaternary ammonium salts is already known. According to the GHS (Globally Harmonized System) hazard statements the two quaternary ammonium salts are: H410 - "Very toxic to aquatic life with long lasting effects". Therefore the author's conclusion stating that this has the potential to be used as an alternative to common chemical fungicides for soil disinfection seams pretty unthinkable.
Anyway is better for the authors to take in consideration some minor adjustments:

  • Line 13 - 14 The scientific name for the two Fusariums needs to be stated only fully.

Response: Thank you for your suggestion. The two fungal abbreviations have been removed. Line 13 – 14.

  • Line 32 Is better to also state the scientific name of the ginger plant.

Response: Thank you for your advice. The scientific name of ginger has been noted in brackets. Line 31.

  • Line 37 - 38 The scientific name for the two Fusariums needs to be stated only fully.

Response: Thank you for your suggestion. The two fungal abbreviations have been removed. Line 36 – 37.

  • Shortly after line 117 I suggest to make the unit measure in the formula in mm as in the text.

Response: Thank you for your advice. The unit measure in the formula has been changed to "mm". Line 117.

  • Line 176 Maybe is better to explain the acronym the first citation.

Response: Thank you for pointing out the problem. We've put the acronym explanation on the first citation. Line 175 – 176.

  • Line 373 Explain the acronym QAC.

Response: Thank you for your suggestion. This phrase occurs only once in the section, so we have eliminated the use of abbreviations and " QAC " has been changed to " quaternary ammonium salts ". Line 376.

  • In References the reference 11 and 12 are not cited anywhere in the text.

Response: Thank you for your query. These two references were used to explain the toxic hazards of methyl bromide. Lines 39 – 43.

This manuscript is a resubmission of an earlier submission. The following is a list of the peer review reports and author responses from that submission.

Round 1

Reviewer 1 Report

Comments and Suggestions for Authors

Dear authors, I have read and evaluated this study with interest. I can't shake the impression that your professional focus is more in another scientific field. And if I'm wrong, then how is it possible that you don't distinguish, for example, bacteria and fungi throughout the text. However, I would like to help the acceptance of this article. I propose to react to the essential suggestions that I have mentioned below.

1) Introduction:L51 etc.  first you write about benzylsulfonium chloride as (BEC) then you mention benzylammonium chloride in the objectives of the study. It is very confusing what substance it really is. It is obvious from the title and abstract that it should be benzylammonium. Please correct it.

2) Material and methods: L73: Purity and etc of BEC are missing.

3) L80: Study is focused on pathogenic bacteria? Correct it in the whole text. L81; L181 etc... 

4) 2.4. centrifuge tubes: provide the volume of the tubes used.

5) 2.5. Plate inhibition replace with Growth inhibition or Colony growth inhibition.

6) term drugs is not suitable for the purpose of BEC and BAC, in this context. Better would be replace term Drug with Compound. The term drug refers more to substances in pharmacology.

7) 2.6 inactivation procedure replace with Spore inactivation assay and give the information about the tubes (ml, origin etc). 

8) in 2.6 you state that Supernatant was removed after centrifugation and resuspended in PBS...to get rid of a desinfectants. Why? It sounds like you are working then only with this supernatant. I assume you meant sediment with spores. Please edit this paragraph.

9) 2.7. Influencing factors would be better to merge with 2.6 (Spore inactivation assay or maybe experiment)

10) 2.8 inactivation mechanism - it makes no sense at all when instead of mechanisms you describe methods and techniques for observing and determining spore inactivation. 2.8.1 - 2.8.3 are simply technics used for your assay or experiment, not mechanisms.

11) 2.2 L80 Isolation and characterization of pathogenic bacteria? Why the bacteria are still mentioned there? Probably you should change it to fungi. 

12) 2.2. Provide more information about the isolation method, how were the cultures purified? was any selective medium used? Was at least PDA with added antibiotics used?

13) 2.2. An information about the DNA isolation method must be added, for example: what kit or method was used, from which company. Which ITS primers were used. Also, provide information on what phylogenetic analysis was used to determine fusaria species. Software, Database etc.

14) 3. Results and Discussion

21 samples from 9 areas should result in you getting more and at least 9 isolates and thus pathogen strains. If these are the ones you list in the dendrogram, please highlight it. If not, state which two strains or isolates of F. oxysporum and F. solani were selected for your study.

15) 3.2 bactericidal effect of chloropicrin L207

it's nonsense if you didn't do anything with the bacteria. Change it to an antifungal.

16) L225 figure S1 you have mentioned is not possible to see in the main text and would be good to highlites that it is in suplementary materials only.

17) L262 two fungicides replace with BAC and BEC.

18) L 264 both fungicides replace with Both antifungal compounds.

19) Figure 7: how to explain the apparent slight (approx. 10%) spore activation in the case of F. oxysporum when 10% BAC is used over time? It would be good to supplement it with a control variant with only water. The explnation of the shape of the curve over time would be good to mention in the discussion. 

20) L326 both fungicides replace with both compounds.

21) The high stability of BAC and BEC is mentioned in the introduction. this is sometimes considered environmentally disadvantageous because of the potential effect on beneficial non-target organisms. How stable are these salts? what is the fate of these compounds in soil? How are they degrades in the soil? It would be good to mentione it in the discussion. Why their use is more advantageous than the existing chlorpicrin. In some parts of the world BAC and BEC are considered problematic and in some places even prohibited, this needs to be justified in the discussion.

Reviewer 2 Report

Comments and Suggestions for Authors

Comments for Authors

The paper discusses the results obtained from a study to evaluate the efficiency of two ammonium salts to control two fungal pathogens. Although the study is investigating an interesting topic, the manuscript has to be improved significantly before publishing.

Firstly, it seems the authors are confused about the pathogens that they are working with, which is not acceptable. Fusarium solani and F. oxysporum are fungal pathogens. Researchers who work with pathogens are supposed to know whether the pathogen is a fungus or bacteria as research methodologies to use with two groups are different from each other. It is a fundamental rule in pathology to identify the pathogen before setting up research methods.

Authors should not use different names for the chemicals that they tested. It is very confusing as they use different terms for these chemicals, instead of the name. For example, authors alternatively use ‘bactericides’,  ‘fungicides’, or  ‘drugs’  instead of the name of the chemicals.

In the manuscript, all the methods have not been explained fully and some important information is missing and therefore difficult to follow. This section is expected to be comprehensive. Please describe the processes succinctly and include all the important information in the method section to enable anyone to reproduce your results if needed.  Some methods described are more suitable for bacteria and should not be used with fungal pathogens. For example, bacterial cells multiply by budding and will increase the number of cells in a suspension with time. Fungal spores do not multiply so the spore numbers will remain the same, but if they are viable, will germinate to produce germ tubes. The efficacy of a fungicide is usually calculated by the % germination reduction. If authors used these methods for a specific reason, please give a valid reason.  

I  do not think that the authors need to have a separate subheading to give names and purchasing information for the chemicals/reagents. Otherwise, they could mention this information within brackets, when you first mention the chemical or reagent. Example: PDB (Difco Laboratories, Detroit, MI, USA)

The data have not been analyzed statistically before concluding results. I strongly advise authors to use an appropriate nonparametric test to analyze data.

I also suggest authors to read about these two pathogens. Fusarium species produce two types of spores, macroconidia, and microconidia. If authors could add information on which type of conidia they used, the paper would be more informative.

Title: Since the study does not give information on effective disease control in ginger plants ( in vivo results), I suggest the title would be

‘ Effective inactivation of spore germination of Fusarium solani and Fusarium oxysporum: ginger soft rot pathogens via’

Or “Benzalkonium chloride and benzylammonium chloride effectively inactivate/reduce spore germination of ginger soft rot pathogens: Fusarium solani and Fusarium oxysporum’

More specific comments are as follows:

Introduction:

Line numbers: 51-56: The study is on two fungal pathogens, Is there information on the effect of these chemicals on fungal pathogens?

Line numbers 60-68:  Do not discuss your results in the introduction. Please move this paragraph to results and discussion.

Materials and methods:

Line 72: Please change the subtitle as the pathogens are NOT BACTERA

Lines 81-92: This is a method to isolate bacteria. The following paragraph from your reference describes the isolation method for fungi.

The pieces were surface sterilized in 70% (v/v) ethanol for 30 s and then rinsed three times with sterile water before drying on sterile filter paper for 1 min. The pieces were then grown on potato dextrose agar (PDA) supplemented with 100 μg L−1 of streptomycin for 7 days at 27 °C for the isolation of fungi. Every single colony of fungi was re-cultured on PDA for the isolation of pure cultures (Reference 31). 

And please explain the reason for using such a high constant temperature (28 °C)  for incubation.

Line 99: Fungal spores are comparatively smaller than blood cells. If you effectively used a blood cell counter to count fungal spores please give the details of the machine (manufacturer and other necessary details).

Lines 100 to 108: The aim of the experiment is not clear here. For accurate results, the spore suspension must be mixed with the soil sample equally. The method does not describe how this was achieved. If the soil samples were incubated for 2 days after adding the spores, they would have germinated by the time of adding the Chloropicrin. The number of replications (2)  is not sufficient enough to draw effective conclusions. Give the reason for excluding Chloropirin from other experiments conducted with BAC and BEC.

Line number 113: Please use the names of the chemicals not “drugs”

Line numbers 116 – 118: “As the control medium mycelium filled the dishes”. Rewrite the sentence for clarity.

Line 120: What is 0.6?

Lines 125-126: Explain how  BAC and BEC were added to the spore suspension (as a liquid?)

Lines 126-130: The method is confusing. What do the authors mean by ‘inactivation reaction’? and inactivation time? What is the disinfectant, BCA or BEC??   Can centrifuging remove BCA or BEC?  If so chemical could not be soluble!

Line 143: What is DMSO?

Line 146: Please be consistent in using “spores’  or ‘conidia’.

Lines 144-146: It is not clear in the method that the Staines were added to the treated spore suspensions or fresh spore suspensions.

Line 153: If you use the term  ‘two fungal spores’, this means you used only two fungal spores for your experiment.

Line 154-155: What is the procedure? This procedure has not been described properly.

Line 157: Did you dilute the sample further? Then, what was the final concentration, the dilution factor?

Line 167: What are the ‘appropriate amounts’?

Line number: 168: Usually, silicon wafers on aluminum stubs are used to mount the specimen.

Line number 180: Please explain what is OD 600.

Line numbers 172-189:  The method used here is not clear. The authors say that they added BAC and BEC to the PDB containing pathogen spores, and then the same spore suspension was mixed with soil. It is not clear here.  

What is the meaning of ‘shaking sterilization’?

Line number: 190-93: Data analysis – Is this equation only for experiment 2.8.4? Some equations are given in the method itself (example subtitle 2.5).

Line number 191: This equation can not be used to calculate the inactivation efficiency of fungal spores. Fungal spores can not multiply like bacteria so the number of spores will remain the same with time. However, if the spores are not affected by the chemical added, they will germinate by producing a germ tube.

Line number 198: What is ‘leaniage method’? This was not described in the method.

Figure 1: The phylogenetic tree should have an out-group. What is 1 and 2??

Line number-205: pathogens can not be isolated from ‘plates’. They should be isolated from diseased tissue.

Line number 206: ‘Experimental conditions’  should be changed to incubation conditions.

Line numbers 221-234: I recommend authors do an appropriate statistical test (Nonparametric test) to find out the differences between treatments to draw valid conclusions.

Line number 240: The meaning of ‘sterilization time’ is not clear.

Line numbers 240-251: The authors draw conclusions without conducting valid statistical tests. Please perform an appropriate statistical analysis before discussing the data.

Line number 287: What does it mean by complete?

Line number 293: Give more information on previous studies for clarity. Were these studies on the same pathogens?

Line numbers 302 -303: “As shown in Figure 9, both F. solani spores and F. oxysporum spores exhibited significant regeneration germination?  potential before inactivation treatments with BCA and ..??

Line number: 315-330: How they collected data to present in Figure 9 or Figure 10 is not clear. Please explain the method clearly.

Comments on the Quality of English Language

Moderate editing of the language will improve the manuscript. 

Reviewer 3 Report

Comments and Suggestions for Authors

The manuscript is significant because it describes inactivationf of ginger soft rot disease caused by Fusarium solani and Fusarium oxysporum using benzalkonium chloride and benzylammonium chloride by several different methods.

My opinion is that the paper could be accepted for publication after minor revision, and adoption of proposed given changes:

Abstract

Line 11: Full and abbreviated Latin names should be placed in the Introduction section also.

Line 20: Insert a few sentences about the methods used.

Line 21 and 23: ...spores of F. solani and F. oxysporum was....

Introduction

Line 30: ....cash and spice crop....

L 34: Put 3 into parentheses.

L 35: Full and abbreviated Latin names should be placed in the Introduction section also.

L 51: Delete the excess point after "agriculture".

L 58: ...spores of F. solani and F. oxysporum was....

L 63: Delete extra space in front of "Inactivation".

L 66: Delete extra space in front of "Measurement".

Materials and Methods

Line 73: Add space in front of "was".

Line 79: Instead of repeating it "was purchased", list the chemicals used in the study with the manufacturer's name in parentheses.

L 81: „Pathogenic fungi“, instead of „bacteria“.

L 82: To mention the previous method, add reference.

Line 91: Sentence "Eventually ... ... isolated" should be deleted, as  it is result and placed in Results section.

Lines 91, 108, 119, 130, 136: Add references for the methods used.

L 119, Formula (2): "Mycelial" instead of "ycelial".

L 139: Add the model and producer of the microscope.

L 153: .".. spores of two fungal species..."

Results and Discussion

L 196: ..."pathogenic fungi" instead of "bacteria".

Line 196: "...pathogenic fungi from ginger" instead of "causal fungi of ginger"...

Line 197: Do you mean "ginger root"?

Line 213: Mention in the Material and Methods section that Probit analysis was used or that results with LC50 are not published.

Lines 217, 236, 282, 297, 315, 328: Full Latin names should be in the Figures tittle.

 (Figures 3, 4, 6, 7, 8, 9, 10, S1, S3, etc...).

L 231: „Fungicidal“ instead of „bactericidal“.

L 293: Add references of the previous studies mentioned.

L 302: „After inactivation“ instead of “before inactivation“.

Lines 321, 323: Delete extra spaces at L321 and 323.

References

Put Latin names in italic in all references.

Comments on the Quality of English Language

The writing style needs some improvement.

Author Response

Effective inactivation of ginger soft rot caused by Fusarium solani and Fusarium oxysporum via benzalkonium chloride and benzylammonium chlorideFor research article

Response to Reviewer 2 Comments

1. Summary

2. Questions for General Evaluation

Reviewer’s Evaluation

Response and Revisions

Does the introduction provide sufficient background and include all relevant references?

Yes

Are all the cited references relevant to the research?

Can be improved

This section has been improved.

Is the research design appropriate?

Yes

Are the methods adequately described?

Can be improvede

This section has been improved.

Are the results clearly presented?

Yes

Are the conclusions supported by the results?

Yes

3. Point-by-point response to Comments and Suggestions for Authors

Abstract

Line 11: Full and abbreviated Latin names should be placed in the Introduction section also.

Response: Thank you for your advice. We have added full and abbreviated Latin names to the Introduction section. L 38

Line 20: Insert a few sentences about the methods used.

Response: Thank you for your advice. We inserted a few sentences describing the methods of fungal isolation and identification. L 20-22

Line 21 and 23: ...spores of F. solani and F. oxysporum was....

Response: Thank you for your advice. “F. solani spores and F. oxysporum spores” has been changed to “spores of F. solani and F. oxysporum” in several places in the article. L 24, L 26, L 61, L 113, L 276, L 321, L 334, and L 378.

Introduction

Line 30: ....cash and spice crop....

Response: Thank you for your advice. “cash crop and spice crop” has been changed to “cash and spice crop”.  L33

L 34: Put 3 into parentheses.

Response: Thank you for identifying the issue. Reference "3" has been placed in parentheses. L 37

L 35: Full and abbreviated Latin names should be placed in the Introduction section also.

Response: Thanks for the suggestion. We have added full and abbreviated Latin names to the Introduction section. L 38

L 51: Delete the excess point after "agriculture".

Response: Thank you for pointing out the error. The excess point after "agriculture" has been deleted. L 54

L 58: ...spores of F. solani and F. oxysporum was....

Response: Thank you for your valuable advice! “F. solani spores and F. oxysporum spores” has been changed to “spores of F. solani and F. oxysporum” in several places in the article. L 24, L 26, L 61, L 113, L 276, L 321, L 334, and L 378.

L 63: Delete extra space in front of "Inactivation".

Response: Thank you for pointing out the error. The extra space in front of "Inactivation" has been deleted. L 66.

L 66: Delete extra space in front of "Measurement".

 Response: Thank you for pointing out the error. The extra space in front of " Measurement " has been deleted. L 69.

Materials and Methods

Line 73: Add space in front of "was".

Response: Thank you for pointing out the error. A space has been added before "was". L 75.

Line 79: Instead of repeating it "was purchased", list the chemicals used in the study with the manufacturer's name in parentheses.

Response: Thank you for your advice. The Experimental Reagents section has been re-edited to remove redundancies. L 75-82.

Chloropicrin (99.5%) was purchased from Shandong Anqiu Farmers' Market. BEC (50% in water), 2.5% glutaraldehyde fixative, propidium iodide, and 300 mesh cell fil-tration sieve were purchased from Shanghai Jizi Biochemistry Co. Ltd. BAC (99.7%, Shanghai Dingxian Biotechnology Co., Ltd), potato dextrose agar (PDA, containing 0.1 g/L chloramphenicol, Guangdong Huankai Microbial Technology Co. , Ltd, China), potato dextrose broth (PDB, Beijing Solebao Technology Co., Ltd, China), isoamyl ace-tate (Shanghai McLean Biochemical Technology Co., Ltd, China), PBS phosphate buff-er(Beijing Jianqiang Weiye Technology Co. , China). The experimental water was Wat-son's distilled water.

L 81: „Pathogenic fungi“, instead of „bacteria“.

Response: Thank you for pointing out the error. The error in multiple parts of the article has been changed to “fungi”. L83, 84, 194, and 207.

L 82: To mention the previous method, add reference.

Response: Thank you for your advice. References cited in multiple methods in the article have been added. L 85, 93, 119, 131, 141, and 148.

Line 91: Sentence "Eventually ... ... isolated" should be deleted, as it is result and placed in Results section.

Response: Sentence Thank you for your advice. "Eventually ... ... isolated" has been deleted. L 94.

Lines 91, 108, 119, 130, 136: Add references for the methods used.

Response: Thanks for the suggestion. References cited in multiple methods in the article have been added. L 85, 93, 119, 131, 141, and 148.

L 119, Formula (2): "Mycelial" instead of "ycelial".

Response: Thank you for pointing out the error. “ycelial” in formula (2) has been changed to " Mycelial ". L 132.

L 139: Add the model and producer of the microscope.

Response: Thanks for the suggestion. The model and producer of the microscope have been added as (DMI 8, Leica, Germany). L151.

L 153: .".. spores of two fungal species..."

Response: Thanks for the suggestion. “two fungal spores” has been changed to “spores of two fungal species”. L 165.

Results and Discussion

L 196: ..."pathogenic fungi" instead of "bacteria".

Response: Thank you for pointing out the error. “pathogenic bacteria” has been changed to “pathogenic fungi”. L 207.

Line 196: "...pathogenic fungi from ginger" instead of "causal fungi of ginger"...

Response: Thank you for pointing out the error. “causal fungi of ginger” has been changed to “pathogenic fungi from ginger”. L 208.

Line 197: Do you mean "ginger root"?

Response: Thank you for pointing out the error. “diseased ginger” has been changed to “diseased ginger root”. L 209.

Line 213: Mention in the Material and Methods section that Probit analysis was used or that results with LC50 are not published.

Response: Thank you for your valuable comments! We have incorrectly quoted the LD50 figure for chlorinated bitterness and have changed it. L 225-228.

  1. oxysporum spores from soil fumigated with 100 mg/kg remained viable in plate cultures, a concentration well in excess of that causing sensory irritation and respiratory damage in humans at 8 mg/kg [50].

Lines 217, 236, 282, 297, 315, 328: Full Latin names should be in the Figures tittle.

 (Figures 3, 4, 6, 7, 8, 9, 10, S1, S3, etc...).

Response: Thanks for the suggestion. Latin abbreviations in all figure titles have been changed to full Latin names. L 219, 231, 251, 268, 296, 316, 334, and 347.

L 231: „Fungicidal“ instead of „bactericidal“.

Response: “bactericidal” has been changed to “fungicidal”. L 263.

L 293: Add references of the previous studies mentioned.

Response: Thanks for the suggestion. The section where the method is located has been added. L 312.

“which was consistent with the findings in Section 3.4.”

L 302: „After inactivation“ instead of “before inactivation“.

Response: Thank you for your suggestion. "before and after inactivation" has been changed to "after inactivation". L 316

Lines 321, 323: Delete extra spaces at L321 and 323.

Response: Thank you for pointing out the error. The extra space has been deleted. L 343 344

References

Put Latin names in italic in all references.

Response: Thank you for your suggestion. Latin names in all references are now italicized.

Comments on the Quality of English Language

The writing style needs some improvement.

Response: Thank you for your suggestions on the English writing style of the article! We have made changes to some of the words and phrases to improve the quality of English.